# Influence of Adipokines on Metabolic Dysfunction and Aging

**DOI:** 10.3390/biomedicines12040873

**Published:** 2024-04-15

**Authors:** Seongjoon Park, Isao Shimokawa

**Affiliations:** 1Department of Pathology, Graduate School of Biomedical Sciences, Nagasaki University School of Medicine, 1-12-4 Sakamoto, Nagasaki 852-8523, Japan; shimo@nagasaki-u.ac.jp; 2SAGL, Limited Liability Company, 1-4-34, Kusagae, Chuo-ku, Fukuoka 810-0045, Japan

**Keywords:** adipose tissue, adipokine, metabolic dysfunction, aging, health span

## Abstract

Currently, 30% of the global population is overweight or obese, with projections from the World Obesity Federation suggesting that this figure will surpass 50% by 2035. Adipose tissue dysfunction, a primary characteristic of obesity, is closely associated with an increased risk of metabolic abnormalities, such as hypertension, hyperglycemia, and dyslipidemia, collectively termed metabolic syndrome. In particular, visceral fat accretion is considered as a hallmark of aging and is strongly linked to higher mortality rates in humans. Adipokines, bioactive peptides secreted by adipose tissue, play crucial roles in regulating appetite, satiety, adiposity, and metabolic balance, thereby rendering them key players in alleviating metabolic diseases and potentially extending health span. In this review, we elucidated the role of adipokines in the development of obesity and related metabolic disorders while also exploring the potential of certain adipokines as candidates for longevity interventions.

## 1. Introduction

Obesity is characterized by an excessive accumulation of adipose tissue, and the major causes of metabolic diseases are a disproportionate increase in adipose tissue and insulin resistance. Adequate amounts of adipose tissue are crucial in mammals, serving not only as an energy storage depot but also as an endocrine organ that regulates metabolic function through the secretion of numerous adipokines. However, excessive accumulation of adipose tissue is associated with metabolic dysfunction and increased susceptibility to obesity, diabetes, and cancer [1]. Adipose tissue, commonly known as fat tissue, comprises two main types: white adipose tissue (WAT) and brown adipose tissue (BAT). White adipose tissue acts as an energy reservoir for other organs, whereas brown adipose tissue functions in cold-induced adaptive thermogenesis. Histologically, white adipose tissue is subdivided into two forms, visceral and subcutaneous. The enlargement of visceral adipose tissue, often termed visceral obesity, is strongly linked to inflammation and insulin resistance [2,3]. The expansion of adipose tissue triggers adipocyte death through mechanical and oxidative stresses, as well as hypoxic conditions, leading to the recruitment of proinflammatory macrophages to adipose tissues [4,5,6]. The infiltration of macrophages contributes to adipose tissue dysfunction, inducing inflammation and insulin resistance in individuals with obesity [7]. Obesity-induced cell inflammation accelerates adipose tissue dysfunction, disrupting overall energy homeostasis and increasing susceptibility to age-related diseases [8]. Uncontrolled secretion of adipokines and the senescence-associated secretory phenotype (SASP) resulting from adipose tissue dysfunction are well-recognized features of aging and metabolic diseases [9] (Figure 1). Several epidemiological cohort studies have shown that obesity increases all-cause mortality and reduces life expectancy in humans [10,11,12]. This has been demonstrated by studies using rodents, which show that suppression of obesity through the removal of visceral adipose tissue results in improved insulin action and prolongs lifespan [13,14]. As a depot of energy storage, adipose tissue is now recognized as an endocrine tissue that regulates metabolic diseases, including obesity, and aging through the regulation of various hormones known as adipokines [15]. Since the identification of adiponectin and leptin as representative adipokines that regulate obesity, numerous types of adipokines have been discovered, prompting extensive research into their roles in health and metabolic diseases [16].

Aging is a process in which tissue function gradually deteriorates as the ability to maintain metabolic homeostasis decreases over time, rendering the body vulnerable to external stress and increasing susceptibility to metabolic diseases, such as obesity, type 2 diabetes, and cardiovascular disease [17]. Although the exact mechanism remains unclear, caloric-restriction-induced suppression of oxidative stress and improvement in energy metabolism contribute to lifespan extension and the reduction of age-related metabolic diseases. Furthermore, the enhancement of adipose tissue function through the modulation of adiponectin and fibroblast growth factor 21 (FGF21), a caloric-restriction-induced adipokine, promotes an extension of health span [18,19].

In this review, we elucidate the role of adipokines in regulating metabolic function and discuss their implications for metabolic diseases and health (Table 1).

## 2. Adipokines in the Regulation of Health and Diseases

### 2.1. Adiponectin

Adiponectin, predominantly secreted by adipocytes, is the most abundant adipokine in plasma [20,21]. Adiponectin is associated with insulin secretion and energy expenditure and is negatively correlated with metabolic disease parameters such as body mass index (BMI), as well as glucose, insulin, triglyceride, and visceral fat levels [22]. Adiponectin exhibits antiatherogenic, antidiabetic, anti-inflammatory, and anti-apoptotic effects by inhibiting monocyte adhesion to endothelial cells and suppressing macrophage transformation into foam cells by suppressing the tumor necrosis factor alpha (TNFα)—nuclear factor kappa B (NF-κB) signaling pathway [23]. Circulating adiponectin and TNFα levels are inversely correlated in both lean and obese individuals [24]. Adiponectin was also shown to increase tissue inhibitor of metalloproteinases (TIMP-1) in human monocyte-derived macrophages through IL-10 induction, which plays an important role in the regulation of vascular inflammation [25]. Recently, protective characteristics of adiponectin that preserve β-cell function have also been reported [26]. Additionally, studies using adiponectin transgenic mice identify it as a longevity gene, demonstrating resistance to metabolic effects, improvement in glucose homeostasis, and amelioration of age-related tissue dysfunctions with extension of health span [4]. Elevated adiponectin level is detected in many longevity model mice, such as fat-specific insulin receptor knockout mice, the Ames dwarf mice (df/df), and GHRKO mice [27,28]. Studies in humans also indicate that higher adiponectin levels are considered crucial parameters in caloric-restricted humans and centenarians [29,30,31]. Thus, adiponectin and its related pathways are promising targets for the treatment of metabolic diseases and aging.

### 2.2. Fiboblast Growth Factor 21 (Fgf21)

FGF21, a subfamily of FGF, is produced by the liver, adipose tissue, and skeletal muscle [32]. FGF21 can diffuse away from the tissue of expression and function as an endocrine hormone due to a lack of the FGF heparin-binding domain [33]. FGF21 is predominantly induced in the liver under fasting conditions through peroxisome proliferator-activated receptor alpha (PPARa) [34]. Fgf21 exhibits suppressive effects on hyperglycemia and atherogenic activity [35]. It reduces plasma triglyceride levels by accelerating lipoprotein lipase (LPL)- and cluster of differentiation 36 (CD36)-mediated triglyceride disposal processes in the liver and adipose tissue, along with the thermogenesis-mediated lipid catabolic process in brown adipose tissue [36]. The metabolic effects of FGF21 require co-expression of fibroblast growth factor receptor 1c (FGFR1c) and b-klotho [37,38]. The growth reduction by FGF21 has also been demonstrated by several studies using genetically modified mouse models, in which transgenic mice overexpressing FGF21 are smaller than wild-type mice, and FGF21-knockout mice grow more than wild-type mice under food-restricted conditions [39,40]. Furthermore, the longevity-related effects of FGF21 have been indicated by findings related to the increased lifespan of transgenic Fgf21-overexpressing mice [5]. The specific mechanisms underlying the beneficial effects of FGF21 are unclear but may involve the suppression of the growth hormone (GH)/insulin-like growth factor 1 (IGF-1) signaling axis in the liver, along with adiponectin [5,41,42].

### 2.3. Adipsin

Adipsin, the first adipokine discovered in 1987 [43], is predominantly expressed in white adipose tissue, especially in subcutaneous adipose tissue, and is implicated in the development of obesity and type 2 diabetes [44]. Adipsin is mainly produced by adipocytes via PPARγ [45,46], and its circulating levels are decreased in obese mice [44]. Depletion of adipsin induces glucose intolerance resulting from beta-cell failure, whereas replenishment of adipsin decreases blood glucose levels through appropriate insulin secretion in obese mice, highlighting its crucial role in maintaining glucose homeostasis and beta-cell function [47]. However, a recent study showed that mice lacking adipsin suppress the expansion of marrow adipose tissue (MAT), thereby inhibiting bone loss during obesity and aging [48], indicating that adipsin has a positive association with glucose-insulin homeostasis but has a negative association with bone remodeling.

### 2.4. Apelin

Apelin, a regulatory peptide identified as an endogenous ligand of the G protein-coupled receptor (APJ) [49], is widely distributed in the body, including adipose tissue (mainly adipocytes), the central nervous system, the heart, skeletal muscle, and the stomach [50]. Apelin is cleaved by the cells to produce endogenous peptides such as apelin-12, -13, -17, and -36 [51]. Apelin levels in the serum and adipose tissue are upregulated in obese and insulin-resistant mice, and apelin contributes to the regulation of food intake, cell proliferation, blood pressure, lipolysis, and glucose metabolism [49,52,53]. Apelin was shown to suppress insulin resistance by increasing AMP-activated protein kinase (AMPK)-mediated glucose utilization and stimulating glucose transporter (Glut) 4, involved in the PI3K and Akt signaling pathways. Comprehensive research using apelin-knockout mice has shown it to induce hyperinsulinemia and insulin resistance [54,55], while studies using apelin treatment mice have shown beneficial functions in obesity and insulin resistance, indicating that apelin could serve as a therapeutic target for treating obesity and related diseases [56,57]. Additionally, apelin has been shown to exert protective effects against bone metabolism through proliferation of osteoblasts via the APJ/PI3k/Akt pathway. Apelin and APJ are also expressed in vascular smooth muscle cells, endothelial cells, and myocardial cells, and low apelin levels are reported in patients with heart failure, suggesting that they are involved in the myocardial response to infarction and ischemia [58,59,60]. The function of apelin in aging has been reported to be that it regulates inflammation, apoptosis, and oxidative stress, which increases during the aging process [61].

### 2.5. Omentin

Omentin, also known as intelectin-1, is primarily produced by visceral adipose tissue and is another potential regulator of insulin sensitivity [62,63]. Encoded by omentin-1 and omentin-2 genes, particularly omentin-1, the main circulating form is positively correlated with adiponectin and high-density lipoprotein levels and negatively correlated with BMI, insulin resistance, triglycerides, and leptin levels [64,65]. Omentin exerts anti-inflammatory effects by inhibiting TNF-α-induced cyclooxygenase-2 (COX-2) expression and Jun N-terminal kinase signaling via activation of AMPK and endothelial nitric oxide synthase [66,67]. Omentin also enhances the stability of atherosclerotic plaque by modulating macrophage viability and inflammation [68]. Furthermore, several studies have indicated a decrease in omentin levels in obesity, cancer, and various cardiovascular diseases, including carotid atherosclerosis, coronary artery disease, heart failure, and dilated cardiomyopathy [64,69,70].

### 2.6. Annexin

Annexins constitute 12 structurally related Ca^2+^- and membrane-binding proteins (AnxA1-AnxA111 and AnxA13) [71,72]. ANXA1, the first identified and extensively studied member of the annexin family, is abundantly expressed in macrophages and neutrophils, and its expression is increased in obesity [73,74]. ANXA1 has been proposed as an anti-inflammatory protein that regulates peripheral leukocyte migration and is a promoter of macrophage phagocytosis in apoptotic neutrophils [75]. Furthermore, ANXA1 is involved in protecting hepatic function, as well as regulating various adipose tissue functions, including those related to inflammation, lipolysis, lipogenesis, and adiposity [76,77]. A study employing ANXA1-knockout mice revealed accelerated hepatic inflammation and fibrosis, elevated glucose and insulin levels, increased adiposity, and decreased insulin sensitivity, emphasizing the significance of ANXA1 in these processes [73,76]. ANXA1 has also been reported to exert a protective effect in resolving inflammation and maintaining vascular homeostasis [78].

Overall, the studies on annexin 1 mentioned above show that annexin 1 alleviates metabolic and vascular diseases by regulating adipose tissue metabolism and inflammation.

### 2.7. Neuregulin (Nrg)

Neuregulin, a member of the epidermal growth factor (EGF) family of extracellular ligands, comprises four isoforms, namely Nrg1–4 [79]. Nrg1, an extensively studied and ubiquitously expressed protein in endothelial and mesenchymal cells, is implicated in cell proliferation, survival, migration, and differentiation [80]. Rodent studies have shown that Nrg1 reduces hepatic glucose production via the ErbB3/Akt signaling pathway [81], indicating its involvement in the regulation of glucose homeostasis. Research across various rodent species and naked mole rats, characterized by longevity, has demonstrated higher Nrg1 levels in longer-lived rodents, suggesting a potential link between Nrg1 and the longevity pathway.

Nrg4, secreted by white and brown adipose tissues, is involved in the regulation of tissue development and tumorigenesis and has been recently discovered in comparison to other adipokines [82]. Nrg4 expression in adipose tissue is lower in obese individuals but increases upon exposure to cold temperatures or epinephrine, suggesting that Nrg4 is involved in the regulation of adipose tissue innervation. However, a study using Nrg4-knockout mice showed insulin resistance under a high-fat diet, but the rectal temperature and expression of the representative thermogenic genes UCP1 and Dio2 did not change under cold stimulation, indicating that Nrg4 is not directly linked to thermogenesis in brown adipose tissue [82]. A binding assay to identify the target of Nrg4 showed that Nrg4 specifically binds to the liver and improves diet-induced fatty liver disease by attenuating the hepatic lipogenic pathway [82], suggesting that circulating Nrg4 from adipose tissue ameliorates the severity of fatty liver and insulin resistance by modulating hepatic lipogenesis.

### 2.8. Leptin

Leptin, a 16 kDa adipocyte-derived adipokine, is considered a potential marker for obesity-related complications such as atherosclerosis [83] and neuropathy [84]. The obese phenotype observed in ob/ob mice, characterized by leptin deletion, is associated with hyperglycemia and insulin resistance [85]. Circulating leptin levels are positively correlated with BMI and adiposity, and their levels are significantly higher in obesity [86,87]. Leptin regulates appetite and energy expenditure by inhibiting neuropeptide Y (NPY), pro-opiomelanocortin (POMC), and corticoliberin (CRH) [88,89] and enhances insulin sensitivity by increasing glucose uptake and oxidation in skeletal muscle and free fatty acid oxidation [90]. However, leptin fails to inhibit appetite and body weight in obese people due to leptin resistance, suggesting that improvement of leptin sensitivity is important for clinical treatment [91]. Owing to the opposing effects of leptin and adiponectin on inflammation and insulin resistance, their ratio has been proposed as a marker of adipose tissue dysfunction [92]. Furthermore, leptin plays a pivotal role in the regulation of satiety, fertility, puberty, activity, and fetal growth [93,94]. The function of leptin in aging has been reported to be that it enhances the vascular aging by calcification of vascular cells [95].

### 2.9. Resistin

Resistin was originally discovered as an adipocyte-specific hormone in rodents and was named for its ability to resist insulin action [96]. This leads to the development of obesity and type 2 diabetes mellitus [96]. Unlike rodents, human resistin is mainly expressed in peripheral blood mononuclear cells, bone marrow cells, and macrophages other than adipocytes, and it accelerates the inflammatory response via NF-κB-mediated activation of TNFa, IL16, and MCP1, classifying it as a proinflammatory molecule [97,98,99,100]. The functional variabilities between mice and human resistin may result from the difference in the 3′ introns. Mouse resistin carries a very large intron in the 3′ UTR, which has a number of regulatory sequences, including the PPAR/RXR binding element [101]. Moreover, resistin levels increase in patients with metabolic syndrome, including obese individuals, and positively correlate with BMI and white adipose tissue mass [102,103]. Resistin has also been involved in age and age-related diseases [104] and is a risk factor for all-cause mortality in elderly people, based on the Finnish cohort study [105]. The inhibition of AMPK and SIRT1, which are crucial in cellular senescence and metabolic regulation, by resistin has been proposed as a conserved mechanism underlying cellular senescence and aging in both humans and mice, despite species diversity [106,107]. These observations suggest that resistin plays a central role not only in the development of insulin resistance and inflammation but also in age and age-related diseases.

### 2.10. Visfatin/NAMPT

Visfatin, also known as nicotinamide phosphate ribosyltransferase (NAMPT), is a product of the pancreatic beta-cell growth factor (PBEF) gene and is predominantly produced by adipocytes and macrophages in visceral adipose tissue [108,109]. The insulin-mimetic activity of visfatin by binding to the insulin receptor, but in a distinct site from insulin, was first demonstrated by Fukuhara et al. [109]. Elevated in obesity, insulin resistance, and type 2 diabetes, visfatin stimulates triacylglycerol synthesis and storage in adipose tissue through activation of glucose uptake and lipogenesis [109,110]. Additionally, visfatin induces the expression of proinflammatory cytokines, such as TNFa, IL1b, and IL6, thereby increasing monocyte–endothelial cell adhesion [111]. The role of visfatin in health is controversial and remains unclear. Despite its positive association with obesity under calorie excess [111], visfatin reportedly improves longevity by enhancing cell survival and SIRT1 activity, as well as through its neuroprotective effects [112,113]. This discrepancy may be attributed to the existence of two distinct forms of visfatin: intravisfatin (iNAMPT), which is positively correlated with obesity under caloric excess, and circulating extravisfatin (eNAMPT), which is associated with anti-aging and longevity effects induced by the suppression of age-related physiological decline through SIRT1-mediated deacetylation of iNAMPT [114,115]. Further studies are required to clarify these controversial findings.

### 2.11. Chemerin

Chemerin, also known as tazarotene-induced gene (TIG)2 and retinoic acid receptor responder (RARRES)2, is primarily secreted by adipose tissue, liver, and immune cells. Chemerin regulates biological processes, such as cell proliferation and differentiation, angiogenesis, and energy metabolism [116,117,118]. Pro-chemerin is produced by the N-terminal cleavage of pre-pro-chemerin, and chemerin is formed by the C-terminal processing of pro-chemerin [119,120,121]. It was initially reported as a chemotactic factor for immune cells, including dendritic cells and macrophages [119]. Subsequently, chemerin was reported to function as an adipokine related to obesity and inflammation [116]. Chemerin was reported to be elevated in the blood, adipose tissue, and liver from obese rodents [122,123], and it is necessary for adipogenesis due to its interaction with PPARγ [118]. The angiogenic action of chemerin supports the notion that chemerin enhances adipose tissue growth by inducing angiogenesis and vascularization [124]. Although chemerin is positively correlated with inflammation and obesity [116], its role, including processing, isoforms, and biological activity in obesity, remains unclear [125]. Furthermore, chemerin acts as a ligand activator of chemokine-like receptor (CMKLR)-1 and as an initiator of innate and adaptive immune responses [126]. Human studies involving obesity and centenarians have suggested that serum chemerin levels are negatively associated with successful aging and health [127,128].

### 2.12. Vaspin

Visceral adipose tissue-derived serpin (vaspin), a member of the serine protease inhibitor family, is highly expressed in adipose tissue [129]. Elevated vaspin levels in rodents and humans are correlated with obesity [129,130,131]. Vaspin regulates insulin sensitivity, preadipocyte differentiation, and angiogenesis [132]. The role of vaspin in suppressing inflammation and insulin resistance was also demonstrated by a study in which administration of vaspin improved glucose tolerance and insulin sensitivity, inhibited proinflammatory cytokines, such as TNFa, resistin, and leptin, and increased levels of adiponectin and GLUT4 in the white adipose tissue of obese mice [129]. The elevated adipocytes differentiation by vaspin was also proven by a study using 3T3-L1 adipocytes in which treatment with vaspin increased expression of PPARg, CEBPa, and CEBPb [133]. Furthermore, vaspin promotes glucose uptake to skeletal muscle through GLTU4 in obese humans [134]. These findings indicate that vaspin appears to be a useful therapeutic candidate for metabolic diseases, including obesity and type 2 diabetes mellitus.

### 2.13. Lipocalin-2

Lipocalin-2 (LCN2), also known as neutrophil gelatinase-associated lipocalin, was initially identified as a secretory protein mainly produced by activated astrocytes and microglia [135]. LCN2 is considered an important regulator of the immune response caused by high expression during infection [136]. Recently, LCN2 has been reported as a new adipokine that is upregulated in obese mice and humans [137,138]. The critical role of LCN2 in metabolic disorders has been demonstrated by studies using LCN2-knockout mice that gained more weight and developed dyslipidemia and insulin resistance [139,140,141]. LCN2 is also involved in the regulation of TNF-mediated inflammatory signaling [142,143] and thermogenesis [144]. Furthermore, LCN2 is secreted by the bone marrow, inhibits food intake in a melanocortin 4 receptor (MC4R)-dependent manner, maintains glucose homeostasis by increasing insulin secretion, and improves glucose tolerance [145]. LCN2 is increased in aging-related brain diseases such as Alzheimer’s disease, Parkinson’s disease, and vascular dementia and is reported to play a role in suppressing neurodegenerative processes [146]. These data demonstrate that Lcn2 is regulated by metabolic stress and inflammatory and nutrient signals, suggesting a pivotal role for LCN2 in metabolic disorders and inflammatory diseases.

### 2.14. RBP4

Retinol-binding protein 4 (RBP4), a member of the lipocalin protein family, is the only known specific transport protein responsible for delivering retinol (vitamin A) in the circulatory system [147,148,149]. RBP4 is primarily produced by the liver and adipose tissue, and its expression is elevated in insulin-resistant mice and humans with obesity and type 2 diabetes [150,151,152]. A study using genetically modified mice showed that transgenic overexpression of RBP4 caused insulin resistance, whereas genetic deletion of RBP4 enhanced insulin sensitivity [152]. The mechanisms of RBP4 involved in insulin sensitization have been demonstrated to be that it alters insulin sensitivity by affecting insulin signaling in muscles through modulation of tyrosine phosphorylation of IRS1 and PI3K activation [152]. The effects of RBP4 on whole-body glucose metabolism were further proven by studies using muscle-specific RBP4 transgenic mice with glucose intolerance and insulin resistance [153]. Furthermore, the proinflammatory effects of RBP4 have shown that RBP4 primes the NLRP3 inflammasome partially through toll-like receptor 4 (TLR4) and TLR2 in macrophages, which impairs insulin signaling in adipocytes [154,155]. The elevated circulating RBP4 level is also associated with hepatic lipid accumulation and liver steatosis in humans [151,156], The study using the NAFLD model mice further showed that the acceleration of NAFLD in RBP4 transgenic mice was mainly attributed to reduced mitochondrial content and impaired mitochondrial fatty acid β-oxidation [157]. Thus, RBP4 contributes to the development of obesity and its associated diseases, including NAFLD.

Overall, the regulation of RBP4 is a novel therapeutic approach for the deterioration of lipid metabolism.

### 2.15. Fetuin A

Fetuin A (FetA), also known as alpha-2-Heremans-Schmid glycoprotein, is mainly produced by the liver, but it is extensively expressed by multiple tissues, such as adipose tissue, kidneys, the brain, and skin [158,159]. FetA was initially identified as an inhibitor of insulin receptor tyrosine kinase in the muscles and liver [160,161]. As such, FetA, which is involved in the formation of insulin receptors, induces insulin resistance with inflammation, causing metabolic disorders, including type 2 diabetes mellitus and nonalcoholic fatty liver disease [162,163]. The FetA/adiponectin ratio has been proposed as a sensitive indicator for evaluating metabolic syndrome in the elderly [164]. FetA also plays a role in anti-apoptotic action by inhibiting proteolytic cleavage and caspase activity [165]. Furthermore, FetA has been reported to regulate PPARγ phosphorylation at serine 273 through the RAas-MEK-ERK pathway, which inhibits the insulin-sensitizing and anti-inflammatory effects of adiponectin [166,167]. Inhibitory phosphorylation of PPARγ by FetA has been shown to inhibit adipogenesis and impair adipocyte function through crosstalk with CD36 [168,169]. The effects of FetA on brain function, including brain development, neuroprotection, and innate immunity, have also been reported [170,171]. Taken together, FetA may have therapeutic and diagnostic roles in the treatment of metabolic diseases.

## 3. Age-Related Changes in Adipose Tissue and Adipokines

The redistribution of adipose tissue in aging with increased visceral adipose tissue and decreased subcutaneous adipose tissue [172] results in an increase in inflammatory cytokines, which trigger metabolic disorders, such as obesity and type 2 diabetes mellitus [15]. Age-related accumulation of visceral adipose tissue also negatively affects cardiac and brain functions [173,174]. The dysregulation of adipokines caused by abnormal accumulation of visceral fat has been shown in the phenotypes of metabolic diseases, as well as aging. An age-related increase in adipokines (adiponectin, leptin, adipsin, vaspin, resistin, and chemerin) [31,175,176,177,178,179] and age-related decrease in adipokines (FGF21, annexin A1, and visfatin) [78,180,181] have been reported in humans. The construction of an aging adipokine profile based on these human studies of adipokines that changed with aging will contribute to extending health span through regulation of adipose tissue function.

## 4. Adipokines Viewed from Caloric Restriction and Centenarian Studies

Caloric restriction (CR), a decreased calorie intake with maintenance of adequate nutrition, not only reduces the risk of metabolic syndrome, including obesity and diabetes, but also extends the lifespan of numerous species, ranging from yeast to primates [182,183,184]. These beneficial functions of CR have also been gradually proven by human caloric restriction and centenarian studies [185,186,187]. It is known that the beneficial functions of CR in metabolic homeostasis and lifespan extension are due to increased insulin sensitivity and improved adipose tissue function; however, several studies place greater emphasis on the importance of adipose tissue function because even in mTORC2-knockout mice with induced insulin resistance, the beneficial functions of CR are maintained, and some long-lived mice do not show an increase in insulin sensitivity [188,189].

CR improves energy efficiency by increasing the utilization of fat, which has higher calories per gram than carbohydrates, leading to metabolic homeostasis being maintained and lifespan being extended by suppressing adiposity and maintaining adipose tissue function. It has been reported that adipokines secreted by adipose tissue, in particular adiponectin, which positively correlates with CR, and leptin and resistin, which negatively correlate with CR, play an important role in adipose tissue function and other health benefits including maintenance of glucose homeostasis [190,191,192] in humans. Furthermore, in studies of centenarians, CR is established as an eating habit of the majority centenarians, and increased adiponectin levels were considered as their common phenotype [31,193]. Although the detailed mechanism of how improved fat regulation contributes to lifespan extension has not yet been accurately reported, adipokine regulation is likely to be at least partially involved.

## 5. Conclusions

Since the discovery of leptin in 1994, numerous bioactive molecules have been discovered in adipose tissue. Adipokines play crucial roles in glucose homeostasis, fat metabolism, and inflammation. Their discovery emphasized the significance of adipose tissue as a representative endocrine organ that regulates obesity and obesity-related metabolic diseases. In particular, adiponectin and FGF21, which are induced by fasting or caloric restriction, have diverse roles in various tissues controlling metabolic diseases, as well as in delaying aging and promoting longevity (Figure 2). They are anticipated to act as vital mediators for extending health span, which has consistently been a focus area of global research. Notably, studies on centenarians have revealed high adiponectin levels and decreased adiposity, indicating the existence of protective phenotypes associated with longevity and healthy aging in humans. Establishing in-depth research and profiling of adipokines through human studies of caloric restriction and centenarians will help uncover new mechanisms for obesity and anti-aging and develop treatments for them.

## Figures and Tables

**Figure 1 biomedicines-12-00873-f001:**
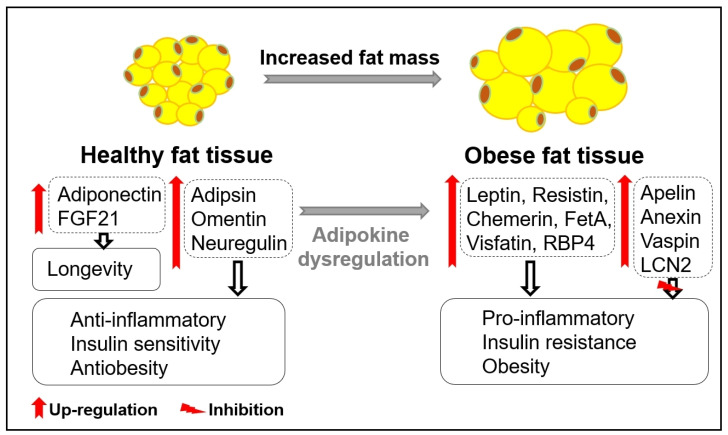
Adipokines in healthy and obese fat tissue. The secretion of beneficial adipokines (adiponectin, FGF21, adipsin, omentin, neuregulin) from healthy fat tissue and detrimental adipokines (leptin, resistin, chemerin, FetA, visfatin, RBP4) from obese fat tissue plays important roles in inflammation, insulin sensitivity, and obesity. Obese-related secretion of other adipokines, such as apelin, annexin, vaspin, and LCN2, plays compensatory roles in inhibiting inflammation, insulin sensitivity, and obesity.

**Figure 2 biomedicines-12-00873-f002:**
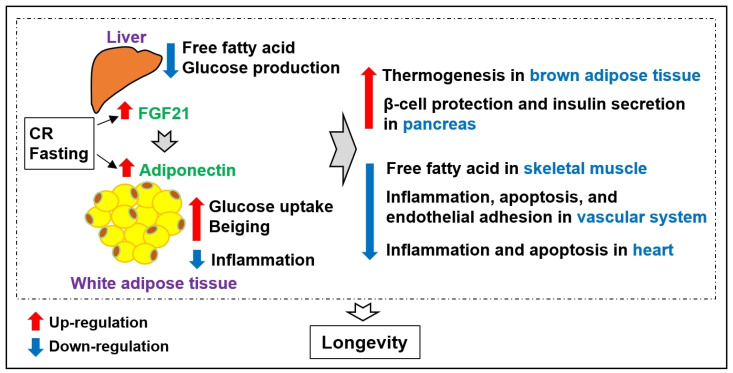
Regulation of physiological functions by adiponectin and FGF21. Adiponectin and FGF21, which are induced by fasting or caloric restriction (CR), have diverse roles in various tissues controlling metabolic diseases and promoting longevity.

**Table 1 biomedicines-12-00873-t001:** Biological effects of adipokines on health and diseases.

Adipokines	Roles
Adiponectin	Improves glucose homeostasis; has antidiabetic, anti-inflammatory, and antiatherogenic effects
FGF21	Improves age-related tissue dysfunctions; extends lifespan;
	positively associated with longevity
Adipsin	Improves glucose tolerance and beta-cell functions;
	stimulates triacylglycerol synthesis and storage in adipose tissue;
	positively associated with longevity; increases cell survival and SIRT1 activity and has neuroprotective effects
Apelin	Regulates food intake; improves glucose disposal
Omentin	Improves insulin sensitivity; has an anti-inflammatory effect
Annexin	Regulates inflammation, lipolysis, lipogenesis, and adiposity
Neuregulin	Regulates cell proliferation, survival, migration, and differentiation;
	reduces hepatic glucose production and lipogenesis;
	stimulates thermogenesis in brown adipose tissue
Leptin	Regulates appetite and energy expenditure;
	negatively associated with longevity
Resistin	Positively associated with obesity and insulin resistance; accelerates inflammation;
	positively correlated with cellular senescence and aging
Visfatin	Stimulates triacylglycerol synthesis and storage in adipose tissue;
	positively associated with longevity; increases cell survival and SIRT1 activity and has neuroprotective effects
Chemerin	Regulates cell proliferation, differentiation, and energy metabolism;
	negatively associated with longevity
Vaspin	Regulates insulin sensitivity, adipocyte differentiation, and angiogenesis; inhibits inflammation
Lipocalin-2	Regulates dyslipidemia and insulin resistance; inhibits inflammation
RBP4	Positively associated with obesity and insulin resistance;
	impairs mitochondrial fatty acid β-oxidation
Fetuin A	Positively associated with insulin resistance and inflammation

The adipokines discussed in this review are summarized in this table.

## Data Availability

No new data were created or analyzed in this study. Data sharing is not applicable to this article.

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
