# Peer review of "Influence of Adipokines on Metabolic Dysfunction and Aging"

_biomedicines, 2024, doi:10.3390/biomedicines12040873_

Round 1

Reviewer 1 Report

Comments and Suggestions for Authors

In this manuscript, the authors summarized and discussed adipokines play crucial roles in regulating obesity, metabolic balance, and their potential to alleviate metabolic diseases and extending health span. On this basis, they further investigated adipose tissue and adipokines in the regulation of health and diseases. This work provides new insight and opinion into the development of adipokines are anticipated to act as vital mediators for extending health span. Overall, the manuscript is well-organized and clearly stated. I would suggest accepting it after the following minor concerns are addressed:

(1)    The authors should reduce keywords in “Section Keywords”.

(2)    The author should modify the Section introduction to highlight the influence of adipokines on obesity and metabolic dysfunction.

(3)    The author should make more detailed comments on Section Figure 1, such as the red lightning part in the figure.

(4)    The author should add more intuitive and summative figures to Section 3.1, 3.2, 3.4 and 3.7, because there are many genes and pathways involved in the description.

(5)    The author should describe the adipokines according to whether they are related to aging. For example, in the description of Section 3.4, 3.5, 3.6, 3.8, 3.12, 3.13, 3.14 and 3.15, it seems that these adipokines have nothing to do with aging.

(6)    The author should delete the description of the influence of adipokines on the systemic energy balance in the Section conclusion, because the relevant experimental results are not cited in the article.

(7)    I suggest adding a summary figure to help people understand the content of the article.

Comments on the Quality of English Language

English in this manuscript is understandable. 

Author Response

Dear Reviewer,

We thank the reviewer for the helpful suggestions. We have followed these suggestions and have extensively revised the manuscript, and we hope that the revised manuscript is now suitable for publication in Biomedicines.

Comments 1: The authors should reduce keywords in “Section Keywords”.

Response: We thank the reviewer for the helpful comment. We have reduced the number of keywords to five.

Comments 2: The author should modify the Section introduction to highlight the influence of adipokines on obesity and metabolic dysfunction.

Response: We thank the reviewer for very helpful suggestions. We combined section 1 and 2 (Introduction, Adipose Tissue in Health and Disease). Additionally, to emphasize the role of adipokines in obesity and metabolic disorders, this content was moved to the front, and the contents of aging and caloric restriction were moved to the back. We also slightly rearranged introduction. We hope that the revised Introduction will further emphasize the impact of adipokines on metabolic dysfunction and provide a better understanding of them.

Comments 3: The author should make more detailed comments on Section Figure 1, such as the red lightning part in the figure.

Response: As the reviewer suggested, a description has been added below Figure 1.

Comments 4: The author should add more intuitive and summative figures to Section 3.1, 3.2, 3.4 and 3.7, because there are many genes and pathways involved in the description.

Response: We agree with the reviewer’s comments. Although it is easy to understand by showing the path and function of each adipokine in pictures, there are some difficulties, such as limitations in the revision period. Instead, we have added a picture summarizing the bioactive functions of adiponectin and FGF21, which are the adipokines most likely to suppress metabolic diseases and extend the lifespan.

Comments 5: The author should describe the adipokines according to whether they are related to aging. For example, in the description of Section 3.4, 3.5, 3.6, 3.8, 3.12, 3.13, 3.14 and 3.15, it seems that these adipokines have nothing to do with aging.

Response: As we mentioned in introduction, aging is a process in which tissue function gradually deteriorates as the ability to maintain metabolic homeostasis and increasing susceptibility to metabolic diseases, such as obesity, type 2 diabetes, and cardiovascular disease. Therefore, the effects of adipokines on metabolic dysfunctions are closely associated with anti-aging interventions. We did not describe separately the effect of all adipokines on metabolic dysfunction (insulin resistance, inflammation, and etc.) and aging, but highlighted the functional role in cardiovascular diseases which are the leading causes of death globally, and longevity in several adipokines. We rearranged the section of introduction and deleted following sentence “In this review, we aimed to elucidate the role of adipokines in aging and metabolic diseases, with a focus on their potential as therapeutic targets for extending health span.” to clarify the purpose of this review. Additionally, we have added description of several adipokines that were conclusively reported about aging (especially oxidative stress, cardiovascular diseases and brain diseases) in several sections (2.4, 2.6, 2.8, 2.13).

Comments 6: The author should delete the description of the influence of adipokines on the systemic energy balance in the Section conclusion, because the relevant experimental results are not cited in the article.

Response: We thank the reviewer for very helpful suggestions. We agree with the reviewer’s point. We have deleted the description of the influence of adipokines on the systemic energy balance in the Section conclusion.

Comments 7: I suggest adding a summary figure to help people understand the content of the article.

Response: As the reviewer suggested, we have added a picture summarizing the bioactive functions of adiponectin and FGF21, which are the adipokines most likely to suppress metabolic diseases and extend the lifespan.

Reviewer 2 Report

Comments and Suggestions for Authors

The present study examined the possibility of certain adipokines as prospective candidates for therapies targeting metabolic diseases and aging. I like providing the following remarks.

1. It established a positive correlation between adipokines. However, a comprehensive investigation of the alterations in adipose tissue and adipokines associated with aging was not carried out. Please enhance this area.

2. The understanding of adipokines in relation to caloric restriction and centenarian research is still uncertain. Kindly restate it with precision.

3. Adipokines, including adiponectin and FGF21, have many functions in metabolic disorders and also contribute to the delay of aging and the promotion of lifespan. This reality will be thoroughly examined in one part, but it will not be discussed in the conclusion.

4. Further information on adipokines would contribute to a better understanding of the new processes, gain insights into the pathophysiology, and facilitate the development of treatments for obesity and aging. It is necessary to provide a clear explanation of this.

5. The present review may be strengthened by its limitations.

Comments on the Quality of English Language

It seems better to check through a professional editing service.

Author Response

Dear Reviewer,

We thank the reviewer for the helpful suggestions. We have followed these suggestions and have extensively revised the manuscript, and we hope that the revised manuscript is now suitable for publication in Biomedicines.

Comments 1: It established a positive correlation between adipokines. However, a comprehensive investigation of the alterations in adipose tissue and adipokines associated with aging was not carried out. Please enhance this area.

Response: We thank the reviewer for the helpful comment. We have already described age-related changes in adipose tissue and adipokines in section 3. There are many research and review papers on the comprehensive investigation of alterations in adipokines and metabolic diseases, but a few papers about adipokines and aging. We hope that our review paper will help in understanding the impact of adipokines not only in metabolic disorders, but also in aging.  

Comments 2: The understanding of adipokines in relation to caloric restriction and centenarian research is still uncertain. Kindly restate it with precision.

Response: We thank the reviewer for the helpful comment. Not all centenarians have dietary restrictions, but especially when examining the results of surveys on centenarians, it is known that their restrictive eating habits contribute to longevity. We have added one more reference about caloric restriction contributing to the longevity of centenarians in section 4. The commonality between changes in adipokines caused by dietary restrictions that induce longevity and changes in adipokines in centenarians suggests that an increase in adiponectin levels may be involved in longevity. In particular, in mouse studies, mice overexpressing adiponectin had an extended lifespan, proving that adiponectin is directly involved in longevity. We describe this in section 2.1 and 4. We hope that our explanation will improve the reviewer’s understanding. 

Comments 3: Adipokines, including adiponectin and FGF21, have many functions in metabolic disorders and also contribute to the delay of aging and the promotion of lifespan. This reality will be thoroughly examined in one part, but it will not be discussed in the conclusion.

Response: We conclude with a brief discussion about Adiponectin and FGF21 in the conclusion. This is concluded as a message of future possibilities and hope because most of the research on Adiponectin and FGF21 is still focused on animal experiments, and human research is limited to the research data on centenarians and caloric restriction mentioned in this review. Additionally, we have added a picture summarizing the bioactive functions of adiponectin and FGF21, which are the adipokines most likely to suppress metabolic diseases and extend the lifespan.

Comments 4: Further information on adipokines would contribute to a better understanding of the new processes, gain insights into the pathophysiology, and facilitate the development of treatments for obesity and aging. It is necessary to provide a clear explanation of this.

Response: We thank the reviewer for very helpful suggestions. We edited the last sentence of the conclusion you pointed out because it seemed a bit difficult to understand. We hope that the revised sentence is easier to understand.

Comments 5: The present review may be strengthened by its limitations.

Comments 6: It seems better to check through a professional editing service.

Response: Our manuscript has already been proofread by a proofreading company, Editage. We attached a certificate of editing from Editage. The content added later was proofread by Paperpal, a proofreading program, owing to the tight revision period.
